# YOUNGER: THE FIRST DATASET FOR ARTIFICIAL INTELLIGENCE-GENERATED NEURAL NETWORK ARCHITECTURE

## ABSTRACT

Designing and optimizing neural network architectures typically require extensive expertise, starting from handcrafted designs followed by manual or automated refinement, which significantly hinders rapid innovation. To address these challenges, Younger is introduced as a comprehensive dataset derived from over 174K real-world models across more than 30 tasks from various public model hubs. After extensive processing and filtering, Younger includes 7,629 unique architectures, each represented as a directed acyclic graph with detailed operator-level information based on ONNX operator definitions, enabling compatibility across different deep learning frameworks. The dataset is designed to support the emerging research area of Artificial Intelligence-Generated Neural Network Architecture (AIGNNA), which aims to automate their generation and refinement. Comprehensive statistical analysis, including architecture component analyses, highlights the diversity and complexity of architectures in Younger, revealing the potential for future research in this domain. Initial experiments, including operator and dataflow predictions, demonstrate the dataset's utility for architecture exploration and evaluation, and highlight its potential as a benchmark for graph neural networks. Furthermore, an online platform ensures continuous maintenance and expansion of the dataset, supporting global researchers in their endeavors. The dataset and source code are publicly available to encourage further research and lower entry barriers in this challenging domain.

## 1 INTRODUCTION

The proliferation of large language models like ChatGPT (OpenAI et al., 2023) has decisively demonstrated the critical importance of large-scale data collection and innovative neural network architecture design in advancing Artificial Intelligence (AI) (Schuhmann et al., 2022; Tang et al., 2024), which has further highlighted the need for more efficient and effective architecture design processes. As a result, in recent years, neural architecture search (NAS) has garnered significant attention and research focus, aiming to address the limitations of manual architecture design in handling increasingly complex AI tasks.

Despite the significant research attention NAS has received, it has not produced revolutionary results. Major architectural shifts, such as the transition from RNNs and CNNs to Transformers, were primarily driven by human designers (Vaswani et al., 2017). NAS frameworks such as DARTS (Liu et al., 2018b) and datasets like NAS-Bench-* (e.g., NAS-Bench-101 (Ying et al., 2019), NAS-Bench-NLP (Klyuchnikov et al., 2022), DeepNets-1M (Knyazev et al., 2021)) have automated the architecture search process. Yet, they remain constrained by predefined search spaces, which refer to specific macro-architecture topologies and a limited set of operator types. This limitation restricts the diversity and novelty of architectures that can be explored, leaving significant room for improvement in neural network architecture design innovation. Instead of creating fundamentally new designs, NAS has primarily focused on pushing model performance within predefined architectural patterns, limiting its ability to generate groundbreaking architectures.

To overcome the limitations of existing NAS frameworks and datasets, and to explore more diverse and innovative architectures, the Younger dataset was developed. Unlike the predefined, constrained

architecture search spaces in NAS datasets such as NAS-Bench-* and NAS frameworks like DARTS, Younger offers a more flexible and diverse exploration space, enabling researchers to break free from rigid macro-architecture topologies and a limited set of operator types.

The Younger dataset is constructed based on the Open Neural Network Exchange (ONNX) operator definitions (Bai et al., 2019), ensuring compatibility across various deep learning frameworks. Derived from approximately 174K real-world models across more than 30 tasks from multiple public repositories (as listed in Table 3), these models are first converted into the ONNX format, then transformed into directed acyclic graphs (DAGs), where nodes represent ONNX operators with detailed configurations and hyperparameters, and edges represent data flows between operators. To ensure uniqueness, isomorphic architectures are filtered out. Additionally, model parameter values are excluded due to concerns about privacy and security. Finally, Younger includes 7,629 distinct neural network architectures and supports all operator types defined by ONNX (about 200 types), striking a balance between the limited operator types in traditional NAS spaces, which typically define only a few operators, and the extensive operator sets in deep learning frameworks like PyTorch, which contains over 2,000 operators, as shown in Table 1. This balance provides a wider range of operator types and data flow configurations than existing NAS spaces while maintaining a manageable exploration space. This flexibility enables researchers to explore novel architectures better suited to the rapidly evolving demands of AI tasks, overcoming the constraints of traditional search spaces.

Extensive statistical analyses were conducted at the operator, component, and architecture levels, validating Younger's capacity to support various design patterns and configurations. These analyses confirmed the dataset's potential for advancing neural architecture research, demonstrating its rich prior knowledge and feasibility in real-world applications. The diversity and complexity of the architectures in Younger provide a robust foundation for exploring new paradigms in neural network design and optimization.

Based on these findings, the Younger dataset has provided a strong foundation for introducing the concept of Artificial Intelligence-Generated Neural Network Architecture (AIGNNA), a powerful approach to automate the generation of neural architectures. This process is characterized by two paradigms: 1) Local, which focuses on fine-tuning and optimizing components of pre-existing architectures by selecting the most suitable operator types and data flows; and 2) Global, which represents the more challenging task of generating entire neural network architectures from scratch, fully automating the design process.

Initial experiments have demonstrated the success of the Local paradigm in optimizing pre-existing architectures, particularly in selecting the most suitable operator types and data flows. However, as the Global paradigm involves generating entire architectures from scratch, it remains an open challenge without readily available methods to automate this process fully. Consequently, while the potential of Younger to support such a paradigm is clear, further research and development are required before comprehensive experiments can be conducted in this area.

To support global collaboration and continuous dataset expansion, Younger is publicly available along with a platform that allows researchers worldwide to upload their models. These models are automatically converted into DAG format and integrated into future releases of Younger, ensuring the dataset remains up-to-date. This open platform lowers the barriers to entry for research in neural architecture generation, empowering researchers worldwide to contribute to and benefit from this evolving field.

## 2 RELATED WORK

### 2.1 ARTIFICIAL INTELLIGENCE-GENERATED NEURAL NETWORK ARCHITECTURE

The design and optimization of neural network architectures have historically been labor-intensive tasks, relying heavily on the intuition and expertise of human researchers. This process has evolved from manual designs, exemplified by early architectures such as AlexNet (Krizhevsky et al., 2012), ResNet (He et al., 2016), LSTM (Hochreiter & Schmidhuber, 1997), and Transformer (Vaswani et al., 2017), to more automated methods employing neural architecture search (NAS) like NASNet (Zoph et al., 2018) and DARTS (Liu et al., 2018b).

Although these manually designed architectures were groundbreaking at the time, they were constrained by the reliance on expert knowledge and required significant time and effort to design. This inefficiency became a growing challenge as the scale and diversity of AI tasks expanded. The emergence of NAS frameworks marked a significant advancement in automating the design process, aiming to improve the efficiency and adaptability of architecture generation.

NAS frameworks such as DARTS (Liu et al., 2018b) and benchmarks like NAS-Bench-101 (Ying et al., 2019) and NATS-Bench Dong et al. (2021) introduced methods that automate the exploration of predefined search spaces, typically consisting of fixed macro-architectures and limited operator types. While these frameworks automate the search for optimal cells (building blocks or micro-architectures) (Zoph et al., 2018; Real et al., 2019; Liu et al., 2018a;b; Pham et al., 2018; Tan & Le, 2019; Klein & Hutter, 2019), they are inherently restricted by these predefined search spaces. As a result, the innovation potential is constrained, limiting the diversity and novelty of the architectures that can be explored.

Addressing these challenges, the Younger dataset and the AIGNNA methodology offer a revolutionary departure from these constraints. By eliminating the need for predefined macro-architectures, Younger allows for a more explorative approach to architecture design, supporting various operator types and data flow configurations, as seen in Table 1. This flexibility facilitates the generation of innovative, customized architectures better suited to specific applications and more adaptable to emerging challenges in neural network design.

Table 1: The difference between Younger and NAS frameworks or datasets

| Dataset / Framework | #op-types | #tasks |
|---|---|---|
| NAS-Bench-101 (Ying et al., 2019) | 3 (CNN) | 1 (Image) |
| NAS-Bench-201 (Dong & Yang, 2019) | 5 (CNN) | 1 (Image) |
| NAS-Bench-NLP (Klyuchnikov et al., 2022) | 6 (RNN) | 1 (Text) |
| NAS-Bench-ASR (Mehrotra et al., 2020) | 6 (CNN) | 1 (Audio) |
| DeepNets-1M (Knyazev et al., 2021) | 15 (CNN) | 1 (Image) |
| NASNet (Zoph et al., 2018) | 13 (CNN) | 1 (Image) |
| DARTS (Liu et al., 2018b) | 4 (RNN) + 7 (CNN) | 2 (Image & Text) |
| **Younger** | $\sim 200$ (ONNX) | 31 (Unlimited) |

## 2.2 Benchmarking Graph Neural Network

Graph neural networks (GNNs) have become a powerful tool for processing graph-structured data across various domains, such as social network analysis, recommendation systems, and molecular chemistry. Traditional benchmark datasets for GNN research, such as Cora, CiteSeer, PubMed (Yang et al., 2016), QM9 (Wu et al., 2018), and ZINC (Gómez-Bombarelli et al., 2018), typically contain graphs with relatively simple and small-scale node and edge structures.

Table 2: The difference between Younger and GNN datasets

| Dataset | #graphs | #nodes | #edges | #node-types |
|---|---|---|---|---|
| Cora (Yang et al., 2016) | 1 | 2,708 | 10,556 | N/A |
| CiteSeer (Yang et al., 2016) | 1 | 3,327 | 9,104 | N/A |
| PubMed (Yang et al., 2016) | 1 | 19,717 | 88,648 | N/A |
| ZINC (Gómez-Bombarelli et al., 2018) | 49,456 | $\sim 23.2$ | $\sim 49.8$ | 10 |
| QM9 (Wu et al., 2018) | 130,831 | $\sim 18.0$ | $\sim 37.3$ | 5 |
| **Younger** | 7,629 | $\sim 1,658$ | $\sim 2,113$ | $\sim 200$ |

Younger introduces a new challenge for GNN research. As shown in Table 2, unlike traditional GNN datasets, Younger features significantly more complex and diverse graph structures, with a notable increase in the number of nodes and edges per graph. Additionally, it supports up to 200 operator types defined by ONNX, far surpassing the node types found in existing datasets. This increased

complexity requires GNNs to handle more extensive and intricate graph topologies, diverse operator types, and data flow configurations.

Despite the increased complexity, Younger maintains a high overall number of DAGs, providing a balanced dataset that offers researchers a broad range of diversity and test scenarios. This balance makes Younger an ideal benchmark for evaluating the scalability, robustness, and generalizability of GNN algorithms while presenting new challenges and opportunities for advancing GNN methodologies.

## 3 DATASET CONSTRUCTION

Collecting real-world neural network architectures is a complex task that demands expertise in deep learning frameworks, especially ONNX (Bai et al., 2019), along with significant computational and human resources. These challenges can be prohibitive for many researchers. A suite of automated tools has been developed to streamline the neural network architecture collection process, facilitate broad support for AIGNNA, and reduce the labor and computational costs associated with data collection.

The dataset construction process involves four key steps: (1) retrieving neural network models, (2) converting models to ONNX format, (3) extracting DAGs from ONNX models, and (4) filtering out isomorphic DAGs to ensure the uniqueness of the architectures. Figure 1 illustrates the entire pipeline. Below is a detailed description of each step:

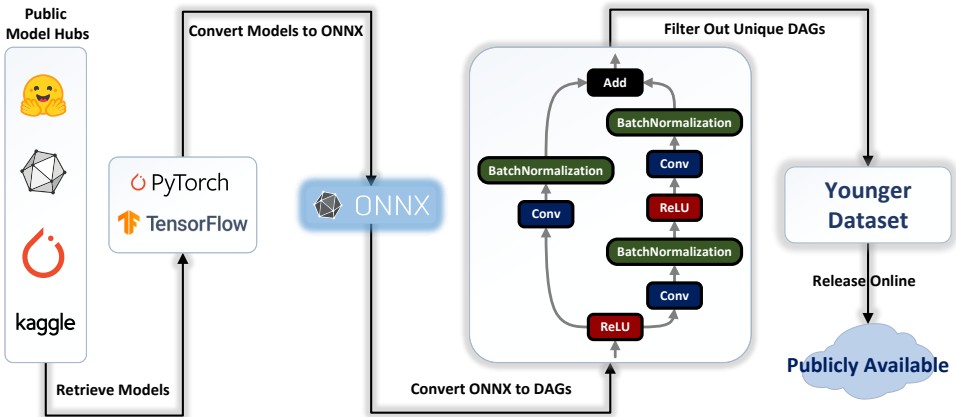

Figure 1: Overview of the construction pipeline

### 3.1 STEP 1: RETRIEVING NEURAL NETWORK MODELS

This study draws from four prominent open-source model repositories to ensure a broad and diverse selection of neural network models. Kaggle Models[1], PyTorch Hub[2], ONNX Model Zoo[3], and Hugging Face Hub[4] are leveraged, collectively encompassing over 30 distinct deep learning tasks. These repositories span diverse deep learning tasks and frameworks, including PyTorch and TensorFlow, ensuring comprehensive coverage of current deep learning models.

To accommodate the rapid growth of repositories like Hugging Face Hub, automated model acquisition tools were implemented to enable continuous updates and ensure timely iteration of the Younger dataset. Although Kaggle Models, PyTorch Hub, and ONNX Model Zoo update frequency is slower, automation tools have also been developed to facilitate efficient model retrieval from these sources. Table 3 provides detailed information about the selected repositories.

---

[1]Kaggle Models: https://www.kaggle.com/models
[2]PyTorch Hub: https://pytorch.org/hub/
[3]ONNX Model Zoo: https://onnx.ai/models/
[4]Hugging Face Hub: https://huggingface.co/models

## 3.2 Step 2: Converting Models to ONNX Format

Different deep learning frameworks define distinct operators, which can lead to increased dataset usage costs and inefficiencies in architecture design when models lack a unified representation. To address this issue, the Open Neural Network Exchange (ONNX) format was adopted as the standard representation for models in the Younger dataset. ONNX provides a standardized set of operators, enabling model exchange and deployment across various deep learning frameworks (such as PyTorch and TensorFlow).

In addition to standardizing operator definitions, ONNX serves as a unified representation, significantly reducing the complexity of neural architecture representation by consolidating operator definitions across frameworks. For instance, ONNX reduces the 2,000+ PyTorch operators to approximately 200 standard operators. Several open-source tools, including Optimum[5] and tf2onnx[6], were utilized to convert models into ONNX format.

## 3.3 Step 3: Extracting DAGs From ONNX models

To address security and privacy concerns, and because neural network architecture design does not require specific parameter values, all parameter data were removed from the ONNX models. Removing parameter data not only addresses security and privacy concerns but also reflects the focus on architecture design, independent of parameterization. Additionally, ONNX models defined in Protocol Buffers[7] format are less suited for direct analysis by standard graph processing tools (e.g., NetworkX (Hagberg et al., 2008)) or deep learning frameworks (e.g., PyTorch Geometric (Fey & Lenssen, 2019)) compared to directed acyclic graphs (DAGs). A tool was developed to convert ONNX models into DAGs to improve compatibility and streamline analysis. This transformation ensures architecture designs can be shared while maintaining parameter privacy and avoiding unnecessary parameter information.

In these DAGs, each operator within a neural network architecture is represented as a node, with detailed information such as the operator type and its attribute definitions recorded. Directed edges represent the data flows between operators, and each node's inflow and outflow order is meticulously documented. The DAGs are represented using the open-source graph library NetworkX, enabling seamless integration with various analysis tools.

Table 3: Statistical information during the construction process of Younger

| Public Model Hubs | Retrievable | Convertable | Retrieved | Converted | Filtered |
|---|---|---|---|---|---|
| Hugging Face Hub | 691K | 325K | 143.5K | 96K | N/A |
| ONNX Model Zoo | 12K | 12K | 12K | 74K | |
| PyTorch Hub | N/A | 121 | 121 | 121 | |
| Kaggle Models | 5K | 4K | 4K | 4K | |
| Total | 743.5K | 341K | 159.5K | 174K | 7,629 |

## 3.4 Step 4: Filtering Out Isomorphic DAGs

Public model hubs often contain many isomorphic neural network architectures, making it necessary to filter these architectures to ensure the uniqueness of each architecture in the dataset. The Weisfeiler-Lehman (WL) graph hash algorithm (Shervashidze et al., 2011) was employed to compute the hash of extracted DAG and identify heterogeneous architectures. The WL algorithm ensures that isomorphic graphs receive identical hash values while heterogeneous graphs are assigned distinct hashes, firmly guaranteeing architectural diversity.

Operator types and their attributes, represented in the nodes, were used as iteration objects within the WL hash algorithm. This process ensures that all architectures in the dataset are heterogeneous, both in terms of hyperparameters and operator types. After applying this filtering method, 7,629 unique neural network architectures were retained from an initial pool of about 174K real-world models.

---

[5]Optimum: `https://github.com/huggingface/optimum`

[6]tf2onnx: `https://github.com/onnx/tensorflow-onnx`

[7]Protocol Buffers `https://protobuf.dev/`

### 3.5 Open-Source Contribution and Global Collaboration

After the whole process, the resulting dataset serves as a foundation for global collaboration and open-source contribution. Overall, the open-source nature of Younger alleviates the need for researchers to invest significant resources in constructing similar datasets. The creation of Younger required substantial computational resources, approximately 8,000 CPU hours, and considerable human effort, including the development of around 24K lines of specialized code. By making both the dataset and its construction methodology open-source, along with accessible interfaces and websites, researchers worldwide can easily contribute to the maintenance and expansion of Younger or build similar datasets. This global collaboration ensures that Younger can continuously evolve to meet the needs of the research community.

When the first version of Younger began construction, there were 743.5K publicly available models, of which 341K could be converted into ONNX format. As of its first release, 174K models were extracted for processing, resulting in 7,629 unique heterogeneous neural network architectures. Despite the vast number of available deep learning models and their rapid growth, less than 1% of these models represent heterogeneous and effective architectures. This notably low proportion of heterogeneous architectures highlights the limitations of current neural network design methods, both manual and NAS-based, in fostering architectural innovation. Younger breaks through these limitations by offering a foundational platform for more flexible and expansive neural architecture design research. It also lays the groundwork for the development of Artificial Intelligence-Generated Neural Network Architecture (AIGNNA), an initiative aimed at exploring new design methods beyond traditional frameworks.

## 4 Experiments

The experiments are divided into two parts: one focuses on the statistical analysis of the Younger dataset, and the other involves an initial exploratory experiment based on the Younger dataset to investigate the proposed AIGNNA.

### 4.1 Experimental Setup

#### 4.1.1 Homogeneous or Heterogeneous?

Neural network operators vary significantly in their attributes. For example, a Convolution (Conv) operator may include attributes such as dilations, kernel shape, and strides, whereas a Batch Normalization operator contains attributes like epsilon and momentum. This diversity poses a critical question in graph-based neural network architecture design: should these architectures be treated as homogeneous or heterogeneous graphs?

In the homogeneous graph approach, all nodes represent the same type (i.e., "operator"), ignoring the specific operator type or its attributes. In contrast, a heterogeneous graph treats the nodes as distinct operator types, capturing the full diversity of operator behaviors and configurations. Although heterogeneous graphs more accurately reflect the complexity of neural network architectures, they introduce additional challenges in analysis and design.

For this study, all architectures in the Younger dataset are treated as homogeneous graphs. This simplification allows the focus to remain on the structural and topological properties of the architectures without introducing excessive variables into the analysis. Future work may explore the treatment of Younger as a heterogeneous graph dataset.

#### 4.1.2 Operator Configurations in the DAGs

Given that the DAG nodes contain discrete information such as operator types and integer attributes, processing node features using conventional approaches can be challenging. To address this, two configurations are explored for handling operator attributes:

**Operator w/o Attributes:** This configuration treats all nodes based solely on their operator types without considering the detailed attribute configurations. This reduces the number of node features to match the size of the ONNX operator set, streamlining the analysis.

**Operator w/ Attributes:** In this approach, operators of the same type but with different attribute configurations are treated as distinct node features. This significantly increases the number of node features, adding complexity to the learning process but more accurately reflecting the detailed structure of the operators. The subsequent experiments will evaluate these two configurations to determine their impact on the learning process.

## 4.2 STATISTICAL ANALYSIS

Statistical analysis is conducted from two perspectives: 1) analyzes lower-dimension statistical information, such as the distribution of the number of nodes in each graph and the operator distribution in Younger. 2) analyzes high-dimension statistical information, including the distribution of three different level granularity: operator, subgraph, and graph.

### 4.2.1 LOW-DIMENSIONAL STATISTICAL INFORMATION

The statistics between Younger and conventional graph datasets are compared. From Table 2 and Figure 2 (a), Younger contains the most extensive distribution of the number of nodes in the graph, ranging from graphs containing only a dozen nodes to graphs containing hundreds of thousands of nodes. In addition, Younger also contains enough graphs compared to most graph datasets, which makes it further challenging to conduct GNNs on Younger. Figure 2 (b) shows Younger's top 30 operators with the highest frequency. The dataset has a great diversity of operator types, including tensor deformations (e.g., Unsequeeze, Reshape), arithmetic operations (e.g., Add, Conv, MatMul), logical operations (e.g., Equal), and quantization (e.g., DynamicQuantizeLinear).

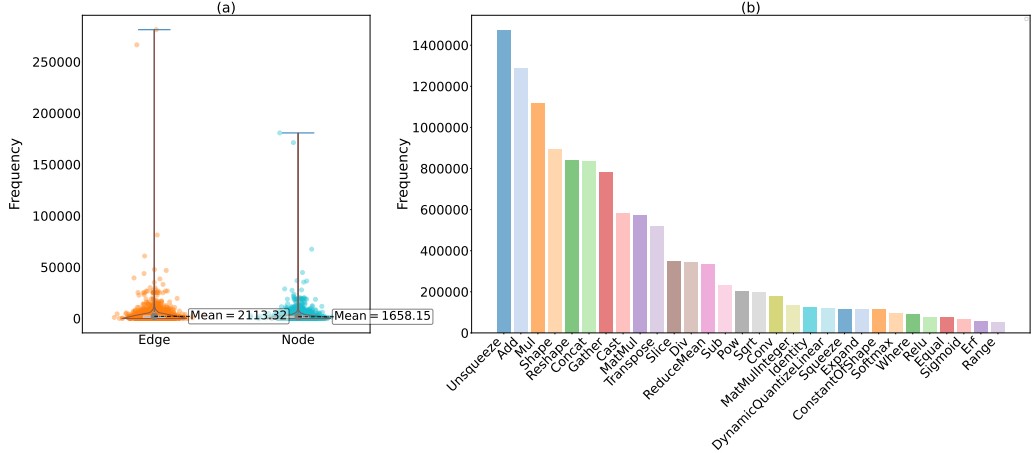

Figure 2: Distribution of #nodes and #edges and top 30 ONNX operators. (a) The distribution of the number of graph nodes and edges in Younger; (b) The top 30 ONNX operators have the highest frequency in Younger.

### 4.2.2 HIGH-DIMENSIONAL STATISTICAL INFORMATION

Due to the nonlinear nature of the graph, embedding techniques were utilized to study the distribution properties of architectures in Younger. Specifically, the GCN Kipf & Welling (2017) network trained in subsection 4.3.2 for operator design is used to obtain the specific embeddings. In Figure 3 and 4, orange dots represent the operators that appear in Younger's top 500 frequencies. After training, GCN gradually extracts the high-frequency operators from the original distribution and aggregates them. This reveals that learning the distribution of long-tailed operators in the dataset is a highly challenging problem. The appendix provides more detailed experimental content.

**Node Embedding** Figure 3-4 show the t-SNE visualization results of node embeddings before and after training from GCN with node features denoted as 'Operator w/ Attributes.' The orange points represent Younger's top 500 most frequently occurring operators. It can be observed that before training, the distribution of node embeddings is relatively concentrated and chaotic. After training,

the distribution of embeddings representing high-frequency nodes selected and other low-frequency nodes from Younger was well distinguished. This indicates an uneven distribution of node quantities among different types, which introduces bias and challenges to the learning process of baseline models.

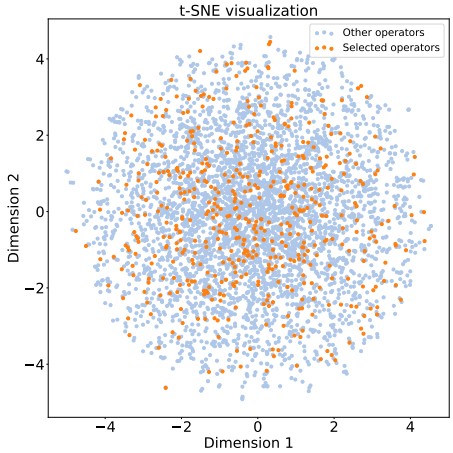

Figure 3: Node embeddings before training

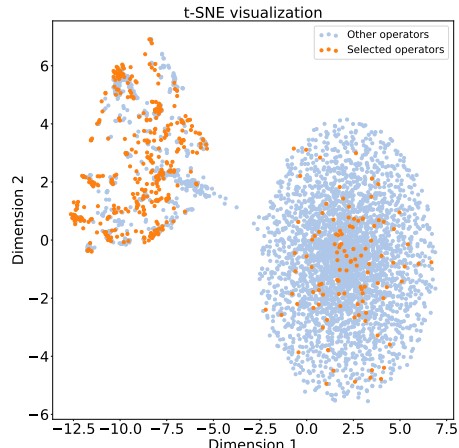

Figure 4: Node embeddings after training

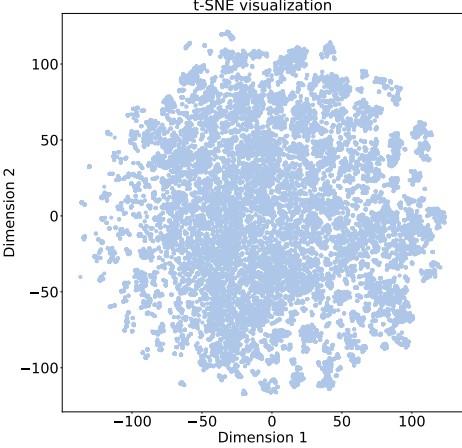

Figure 5: Subgraph embeddings

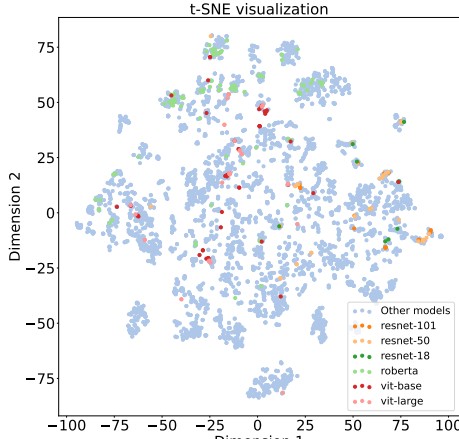

Figure 6: Graph embeddings

**Subgraph Embedding** Figure 5 shows the t-SNE visualization results of all subgraph embeddings under the GCN model. As can be seen, the GCN model have distinguished the embeddings of subgraphs well, but due to data bias, node embeddings were not learned particularly well. Therefore, the model only distinguished the embeddings of subgraphs well in a part of the spatial distribution (the boundary of the space).

**Graph Embedding** Figure 6 shows the t-SNE visualization results of all graph embeddings under the GCN model. The embeddings of several commonly used models in figures are marked in different colors. Several architectures have shown almost similar results. It can be seen that, on the one hand, the embeddings of DAGs based on the same architecture are very close or even overlap in the graph; for example, there are many points of the RoBERTa (Liu et al., 2019) and ViT (Dosovitskiy et al., 2020) architectures, which are Transformer-based (Vaswani et al., 2017) architectures, that are close in distance or overlap. On the other hand, it can be seen that the Younger dataset covers multiple common architectures well, indicating that Younger covers most of the neural network architectures

in the real world. In addition, the same architecture has multiple points of the same color in the figures, indicating that the dataset contains various variants of this type of architecture.

### 4.3 AIGNNA EXPLORATION

Experiments were conducted on the Younger dataset for global and local paradigms to verify the feasibility and effectiveness of the proposed two paradigms for AIGNNA. The results indicate that exploring AIGNNA based on Younger is feasible, demonstrating Younger's potential as a benchmark dataset for graph neural networks.

#### 4.3.1 OVERVIEW OF AIGNNA PARADIGMS

To advance the development of AIGNNA based on the Younger dataset, two paradigms for neural network architecture design are introduced, each tailored to different real-world application scenarios. Figure 7 provides an intuitive visualization of these paradigms.

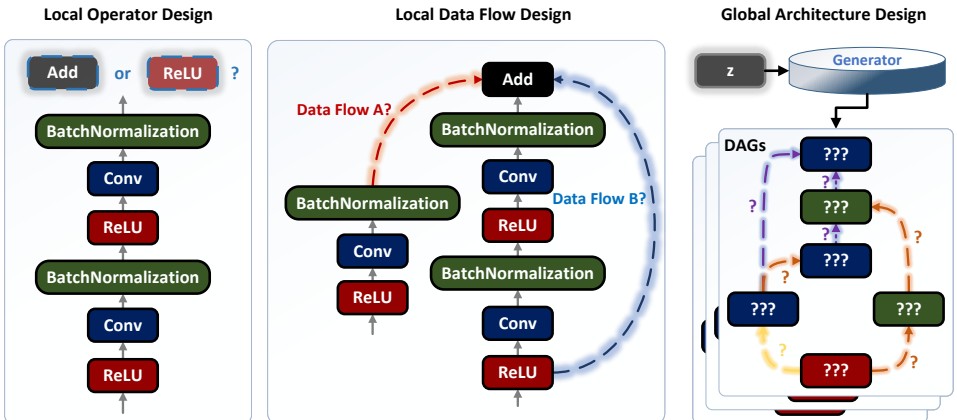

Figure 7: Paradigms of the AIGNNA

**Local: Architecture Refinement In Detail.**The local paradigm addresses the need to fine-tune specific aspects of existing neural network architectures. This approach is divided into operator and data flow designs, as shown in Figure 7. Operator Design involves determining the most suitable type of operator for a given node based on local or global architectural information, as illustrated in the leftward of Figure 7. This design assesses potential replacements for current operators and suggests appropriate operators for new nodes based on neighboring structural information.

The second type, data flow design, evaluates the existence of data flows between operators. This fine-tuning method determines whether a directed edge representing data flow should connect any two nodes, utilizing insights from local and global architectural contexts.

Challenges within the local paradigm arise from the vast diversity of operators and the binary nature of data flow decisions (existing or not). The efficacy of this paradigm is assessed by employing five different graph neural networks as baselines, focusing on operator and data flow design. Operator design presents a greater challenge than data flow design.

**Global: Architecture Design From Scratch.** Designing neural network architectures from scratch is an open and complex challenge. Unlike neural architecture search, which limits the search space to a predefined macro-architecture while optimizing micro-architectural elements for specific performances, the global paradigm seeks to generate comprehensive neural network architectures incorporating detailed operator-level elements from the ground up.

As shown in the rightward flowchart of Figure 7, this generative process is conditioned on specific properties, denoted by $z$ in Figure 7, such as a noise that represents the architecture's intended application or required characteristics. Moreover, the architecture's design objectives are defined by the goals it needs to achieve. Importantly, global paradigms can also iteratively leverage local

paradigms to progressively achieve their comprehensive design objectives. To assess the potential and feasibility of the global paradigm, a robust baseline is implemented for validation.

### 4.3.2 LOCAL PARADIGM

**Data Flow Design:** GCN, GAT (Brody et al., 2022), and GraphSAGE (Hamilton et al., 2017) are employed under the data flow design paradigm to evaluate the effectiveness of neural architecture refinement on the Younger dataset. The results are shown in Table 4. All models have achieved good performance on the Younger dataset, which proves that existing graph neural networks are more suitable for predicting data flows in neural network architectures. Additionally, it can be seen that almost all models perform better without attributes because reducing the number of node features on the graph makes learning them easier.

Table 4: Local paradigm: data flow design

| Model | Operator w/ Attributes | | | Operator w/o Attributes | | |
|---|---|---|---|---|---|---|
| | AUC↑ | F1↑ | AP↑ | AUC↑ | F1↑ | AP↑ |
| GCN | **0.9922** | 0.7881 | **0.9913** | **0.9938** | 0.7791 | **0.9929** |
| GAT | 0.8997 | **0.8079** | 0.8720 | 0.9094 | 0.7964 | 0.8901 |
| SAGE | 0.9169 | 0.8033 | 0.8940 | 0.9252 | **0.8002** | 0.9026 |

**Operator Design:** Five different baselines, GCN, GAT, GAE (Kipf & Welling, 2016), VGAE (Kipf & Welling, 2016), and GraphSAGE, are utilized for ten experiments under the operator design paradigm, as shown in Table 5. Despite the high accuracy achieved by all baselines, the F1 score, Precision, and Recall remain low. This is primarily attributed to the complex graph structures in Younger, which are characterized by many operator types. Among these, multiple kinds of operators infrequently occur, posing challenges to achieving robust multi-classification performance. In experiments without attributes, higher values for F1, Precision, and Recall were observed compared to scenarios with attributes. This result further highlights the inherent complexity of the dataset and its influence on classification performance.

Table 5: Local paradigm: operator design

| Model | Operator w/ Attributes | | | | Operator w/o Attributes | | | |
|---|---|---|---|---|---|---|---|---|
| | ACC↑ | F1↑ | Prec.↑ | Recall↑ | ACC↑ | F1↑ | Prec.↑ | Recall↑ |
| GCN | 0.8684 | 0.1451 | 0.1713 | 0.1466 | 0.8360 | 0.2987 | 0.3657 | 0.2788 |
| GAT | OOM | OOM | OOM | OOM | 0.7139 | 0.2022 | 0.2532 | 0.2039 |
| GAE | **0.9016** | 0.0537 | 0.0728 | 0.0513 | 0.9073 | 0.1745 | 0.2036 | 0.1700 |
| VGAE | 0.8243 | 0.0716 | 0.0891 | 0.0707 | 0.9137 | 0.2207 | 0.2654 | 0.2132 |
| SAGE | 0.8984 | **0.2028** | **0.2383** | **0.1996** | **0.9250** | **0.3646** | **0.4323** | **0.3532** |

### 4.3.3 GLOBAL PARADIGM

In the global paradigm, the graph generation model DiGress, which employs a diffusion model for graph generation, was adopted. Due to computing resource constraints and the fact that some architectures in Younger have node counts reaching hundreds of thousands, only architectures with node counts in the range of $[1, 300]$ were selected for training. The DiGress model achieved a negative log-likelihood of at least 345.4988 on the test set. As the global paradigm presents a highly challenging task, further research in this area is planned for the future.

## 5 CONCLUSION AND FUTURE WORK

This article introduces Younger, a dataset of neural network architectures extracted from real-world models across various public model repositories. This dataset proposes a new challenging field: Artificial Intelligence-Generated Neural Network Architecture (AIGNNA). Two critical challenges regarding neural network architecture design are introduced within this field: the Global Design Paradigm and the Local Design Paradigm. Preliminary experiments have demonstrated the potential and effectiveness of Younger's neural architecture design in this emerging field, encouraging more researchers to explore this promising.

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

# Appendix

## CONTENTS

## A   EXPERIMENTAL DETAILS

This section offers a more detailed examination of the experiments discussed in the main paper. Specifically, it addresses five critical components: Local Data Flow Design and Local Operator Design within the Local Paradigm and Node, Subgraph, and Graph Embedding in the context of Statistical Analysis. It offers a comprehensive introduction and discussion of dataset splits, training details, model selection, results, and analytical insights.

### A.1   LOCAL DATA FLOW DESIGN

#### A.1.1   DATASET SPLITS

Before splitting the dataset, we removed graphs with nodes or edges less than one from the 'Filter' dataset. Subsequently, the dataset was divided into training, validation, and test sets in a ratio of 8:1:1 with a random seed to be set as 1234. To better meet the need for local data flow design, we removed graphs in the validation set and test set with operator type not appearing in the training set to maintain training performance. Ultimately, there were 5994, 690, and 685 unique architectures in training, validation, and test sets for node features denoted as 'Operator w/ Attributes.' For node features denoted as 'Operator w/o Attributes,' there were 5612, 639, and 648 unique architectures in training, validation, and test sets, respectively.

#### A.1.2   BASELINE MODEL CONFIGURATION

The architectures of three baseline models represented by topological diagrams under the local data flow design paradigm are shown in Figure 8 and Table A.1.2 indicates the number of parameters. It is worth mentioning that the outputs of multi-head attention of GAT are averaged instead of concatenated.

Table 6: Number of Parameters of Local Data Flow Design Baseline Models.

| Model | Operator w/ Attributes | Operator w/o Attributes |
|---|---|---|
| | Number of Parameters | Number of Parameters |
| GCN | 5,360,384 | 849,664 |
| GAT | 9,960,192 | 5,449,472 |
| SAGE | 6,015,744 | 1,505,024 |

#### A.1.3   TRAINING CONFIGURATION

In this version, we set the random seed to 12345 and chose Adam as the optimizer for the local data flow design training process. Other hyperparameters were set as shown in Table 7. The experiments for local operator design were conducted on a server running Ubuntu 22.04.1 LTS. It has four identical A800-80GB GPUs and an Intel(R) Xeon(R) Gold 6348 CPU @ 2.60GHz with 112 cores. All the baseline models for data flow design were trained on four identical A800-80GB GPUs.

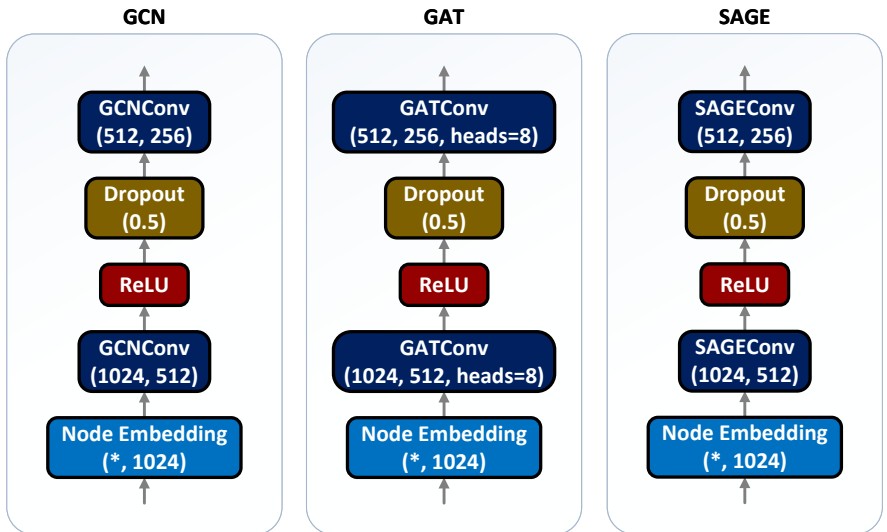

Figure 8: Topological diagram of three baseline models: GCN, GAT, and SAGE .

Table 7: Training Details of Local Data Flow Design. 'LR,' 'WD,' and 'BS' in the header represent Learning Rate, Weight Decay, and Batch Size, respectively

| Model | Operator w/ Attributes | | | Operator w/o Attributes | | |
|---|---|---|---|---|---|---|
| | LR | WD | BS | LR | WD | BS |
| GCN (Kipf & Welling, 2017) | 1e-4 | 5e-5 | 1 | 1e-4 | 5e-5 | 1 |
| GAT (Brody et al., 2022) | 1e-4 | 5e-5 | 1 | 1e-4 | 5e-5 | 1 |
| SAGE (Hamilton et al., 2017) | 1e-4 | 5e-5 | 1 | 1e-4 | 5e-5 | 1 |

### A.1.4 METRICS

**Area under the Receiver Operating Characteristic Curve (AUC):**

$$\text{TPR} = \frac{\text{TP}}{\text{TP} + \text{FN}}, \tag{1}$$

$$\text{FPR} = \frac{\text{FP}}{\text{FP} + \text{TN}}. \tag{2}$$

For the Receiver Operating Characteristic (ROC) Curve, the Y axis represents the true positive rate (TPR) while the X axis represents the false positive rate (FPR). A value of AUC close to 1 represents a better classification prediction performance.

**F1 Score (F1):**

$$\text{F1} = \frac{2 \cdot \text{TP}}{2 \cdot \text{TP} + \text{FP} + \text{FN}}, \tag{3}$$

where TP, FP, and FN represent the number of true positives, false positives, and false negatives, respectively.

**Average Precision (AP):**

$$\text{AP} = \sum_{n=1}^{N} (R_n - R_{n-1}) P_n, \tag{4}$$

where $R$ and $P$ represent the precision and recall, while $n$ denotes the $n$th threshold.

### A.1.5 CHECKPOINT SELECTION

We chose checkpoints to test the performance of baseline models based on the weighted average of all the metrics reported during validation. The weighted averages of AUC, F1, and AP were calculated to measure the performance of baseline models. In this version, all weights are set to be the same.

### A.1.6 Results and Analysis

We set up our configuration as stated in Section A.1.3 and used GCN, GAT, and GraphSAGE for six experiments under the data flow design paradigm on Younger. As shown in Table 8, these three baseline models perform well on all metrics. It is worth noting that GCN outperforms other models in all metrics except F1 Score, regardless of whether the operators have attributes.

Table 8: Local paradigm: data flow design. Bold values represent the best-performing results.

| Model | Operator w/ Attributes | | | Operator w/o Attributes | | |
|---|---|---|---|---|---|---|
| | AUC↑ | F1↑ | AP↑ | AUC↑ | F1↑ | AP↑ |
| GCN (Kipf & Welling, 2017) | **0.9933** | 0.7893 | **0.9924** | **0.9949** | 0.7907 | **0.9942** |
| GAT (Brody et al., 2022) | 0.9195 | **0.8023** | 0.8974 | 0.9133 | 0.7960 | 0.8937 |
| SAGE (Hamilton et al., 2017) | 0.9702 | 0.8005 | 0.9682 | 0.8991 | **0.8053** | 0.8591 |

## A.2 Local Operator Design

### A.2.1 Dataset Splits

Due to the lack of relevant research on extracting building blocks for neural network architecture. Therefore, we performed community detection on all DAGs (Neural Network Architecture) in the 'Filter' dataset to extract the building blocks of the neural network architecture. Through community detection, we can identify the closely connected node sets in the graph to help identify subsets of nodes with specific correlations or functional associations. Although there is no evidence to suggest that the subgraphs extracted by community detection are effective building blocks for neural network architecture, in this paper, it is reasonable to use this method to extract subgraphs for preliminary validation to test the feasibility of Local Operator Design.

We adopt the Clauset Newman Moore Grey modularity maximization method (Clauset et al., 2004) as the community detection algorithm and set it to detect at least one community, the DAG itself. For each community, we simultaneously query its node boundary and label it as the node to be predicted. The community and node boundary form a new subgraph, and the definition of node boundary is shown in Formula A.2.1.

$$\mathcal{B} = \{v | v \in \mathcal{D} - \mathcal{C}, u \in \mathcal{C}, (u, v) \in \mathcal{E}\}, \tag{5}$$

where $\mathcal{D}$, $\mathcal{C}$, and $\mathcal{E}$ represent the node set of DAG and the node set of community and edge set of DAG, respectively, and $(u, v)$ indicates two directed edges $< u, v >$ and $< v, u >$.

Finally, we will deduplicate the subgraphs formed by all community and node boundary pairs, i.e., remove isomorphic subgraphs. Finally, 38,803 and 29,581 non-isomorphic subgraphs were obtained under the configurations of 'Operator w/ Attributes' and 'Operator w/o Attributes', respectively. To obtain the final training, validation, and test sets, we split all non-isomorphic subgraphs in an 8:1:1 ratio. Specifically, under the 'Operator w/ Attribute' configuration, the training, validation, and testing sets contain 31,282, 3,769, and 3,752 subgraphs, respectively, while under the 'Operator w/o Attribute' configuration, they include 23,775, 2,907 and 2,899 subgraphs, respectively.

### A.2.2 Baseline Model Configuration

The architectures of baseline models represented by topological diagrams under the local operator design paradigm are shown in Figure 9 and Table A.2.2 indicates the number of parameters. For experiments with GAE and VGAE under the local operator design paradigm, we first pre-trained the encoders of GAE and VGAE, then trained the linear layers for classification using the output from encoders. For GAT, the outputs of multi-head attention of GAT are averaged instead of concatenated.

### A.2.3 Training Configuration

In this version, we set the random seed to 12345 and chose Adam as the optimizer for the local operator design training process. Other hyperparameters were set as shown in the Table 10. The experiments for local operator design were conducted on a server running Ubuntu 22.04.1 LTS. It has four identical A800-80GB GPUs and an Intel(R) Xeon(R) Gold 6348 CPU @ 2.60GHz with 112

Table 9: Number of Parameters of Local Operator Design Baseline Models.

| Model | Operator w/ Attributes | Operator w/o Attributes |
|---|---|---|
| | Number of Parameters | Number of Parameters |
| GCN | 7,301,433 | 809,145 |
| GAT | 26,852,041 | 5,153,353 |
| SAGE | 10,083,129 | 1,428,153 |
| GAE-Encoder | 6,089,216 | 1,763,840 |
| GAE-Classification | 2,261,817 | 94,905 |
| VGAE-Encoder | 6,614,016 | 2,288,640 |
| VGAE-Classification | 2,261,817 | 94,905 |

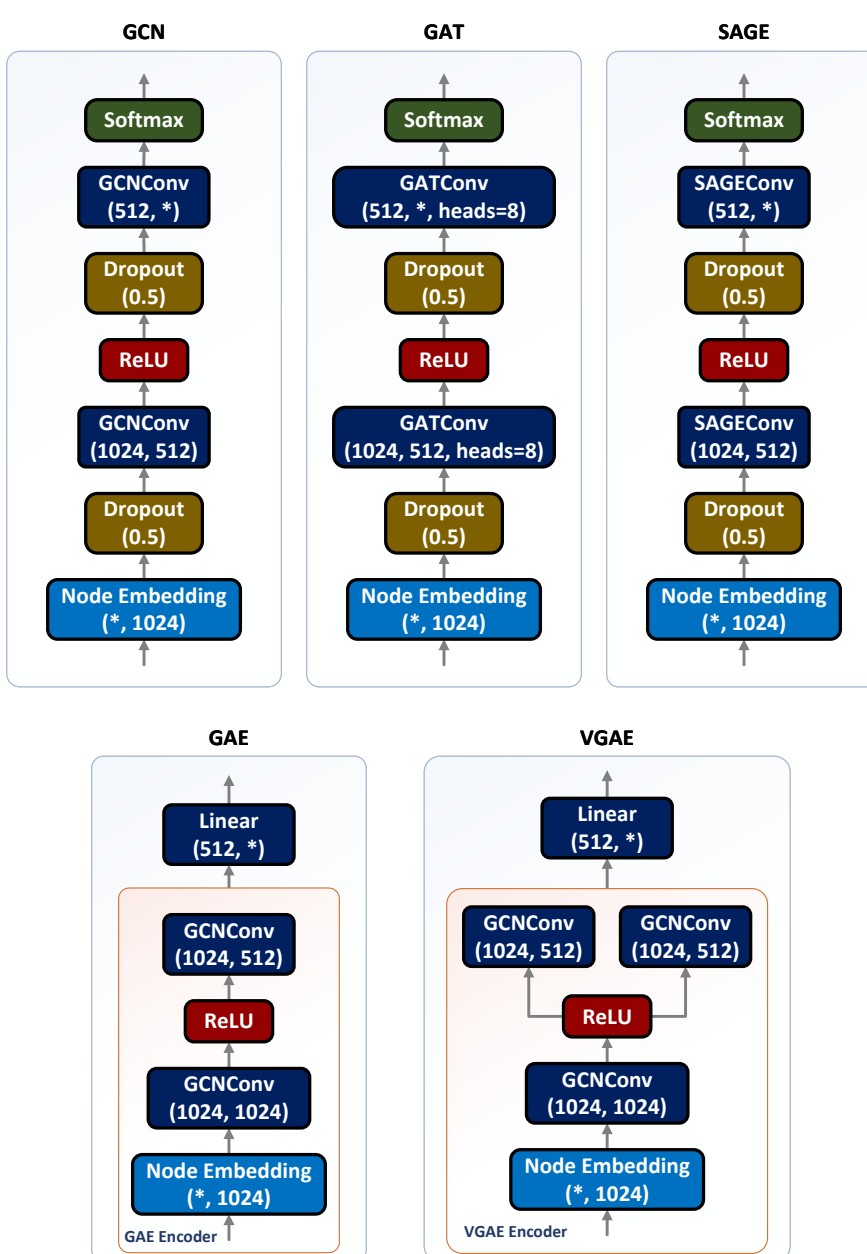

Figure 9: Topological diagram of five baseline models: GCN, GAT, SAGE, GAE, and VGAE .

cores. GAT, GCN, and SAGE were trained on four A800-80GB GPUs, while GAE and VGAE were trained on one A800-80GB GPU.

Table 10: Training Details of Local Operator Design. 'LR,' 'WD,' and 'BS' in the header represent Learning Rate, Weight Decay, and Batch Size, respectively.

| Model | Operator w/ Attributes | | | Operator w/o Attributes | | |
|---|---|---|---|---|---|---|
| | LR | WD | BS | LR | WD | BS |
| GCN | 1e-3 | 5e-5 | 512 | 1e-3 | 5e-5 | 512 |
| GAT | OOM | OOM | OOM | 1e-3 | 5e-5 | 512 |
| SAGE | 1e-3 | 5e-5 | 512 | 1e-3 | 5e-5 | 512 |
| GAE-Encoder | 1e-4 | 5e-5 | 512 | 1e-4 | 5e-5 | 512 |
| GAE-Classification | 1e-3 | 5e-4 | 512 | 1e-3 | 5e-4 | 512 |
| VGAE-Encoder | 1e-4 | 5e-5 | 512 | 1e-4 | 5e-5 | 512 |
| VGAE-Classification | 1e-3 | 5e-4 | 512 | 1e-3 | 5e-4 | 512 |

### A.2.4 METRICS

**Accuracy (ACC)**: The ratio of correctly predicted instances to the total instances.

**F1 Score (F1)**:

$$\text{F1} = \frac{2 \cdot \text{TP}}{2 \cdot \text{TP} + \text{FP} + \text{FN}}, \tag{6}$$

where TP, FP, and FN represent the number of true positives, false positives, and false negatives, respectively.

**Precision (Prec)**:

$$\text{Precision} = \frac{\text{TP}}{\text{TP} + \text{FP}}, \tag{7}$$

where TP and FP represent the number of true positives and false positives.

**Recall**:

$$\text{Recall} = \frac{\text{TP}}{\text{TP} + \text{FN}}, \tag{8}$$

where TP and FN represent the number of true positives and false negatives.

### A.2.5 CHECKPOINT SELECTION

We chose checkpoints to test the performance of baseline models based on the weighted average of ACC, F1 Score, Precision, and Recall reported during validation. In this version, all weights are set to be the same. For the encoder of GAE and VGAE, we chose the checkpoint on training step 4000, whose training loss remained stable.

### A.2.6 RESULTS AND ANALYSIS

We set configuration as stated in Section A.2.3. Baseline models, including GCN, GAT, GAE, VGAE, and SAGE, were used under the operator design paradigm. As shown in Table 11, all baseline models achieve high accuracy but perform poorly in other metrics. The reason can be attributed to the complexity of Younger and further to the complexity of the neural network architectures in the real world. Another reason is that some typical types of operators appear more frequently while others appear less frequently, causing the model to be biased toward predicting the majority of operators. It can be seen that all baseline models in experiments w/o attributes achieve higher F1, Precision, and Recall compared to those in experiments w/ attributes. This indicates that reducing the variety of operators and making their distribution more uniform can improve the multi-classification performance. In addition, among these baseline models, SAGE performs excellently on almost all metrics. Notice that GAT lacks experiments with Operator w/ Attributes due to excessively large parameter counts as shown in Table 9, resulting in out-of-memory issues during execution.

Table 11: Local paradigm: operator design. Bold values represent the best-performing results. 'Prec.' in the header represents Precision.

| Model | Operator w/ Attributes | | | | Operator w/o Attributes | | | |
|---|---|---|---|---|---|---|---|---|
| | ACC↑ | F1↑ | Prec.↑ | Recall↑ | ACC↑ | F1↑ | Prec.↑ | Recall↑ |
| GCN | 0.7454 | 0.1294 | 0.1666 | 0.1323 | 0.7627 | 0.2988 | 0.3750 | 0.2941 |
| GAT | OOM | OOM | OOM | OOM | 0.7163 | 0.2007 | 0.2519 | 0.1979 |
| GAE | 0.8173 | 0.0484 | 0.0658 | 0.0467 | 0.8179 | 0.1514 | 0.1815 | 0.1438 |
| VGAE | **0.8224** | 0.0724 | 0.0924 | 0.0712 | 0.8243 | 0.1969 | 0.2500 | 0.1881 |
| SAGE | 0.8049 | **0.1927** | **0.2385** | **0.1878** | **0.9238** | **0.3477** | **0.4144** | **0.3375** |

## A.3 NODE EMBEDDING

### A.3.1 CHECKPOINT SELECTION

To better illustrate the distribution of operators in Younger in high-dimensional space, we selected checkpoints of baseline models according to the method from section A.2.5 and then extracted the embeddings of operators with attributes and those without attributes from node embedding layer of baseline models. Due to the problem about memory overflow, the visualization of 'Operator w/o Attributes' about GAT is not presented. To compare the training effectiveness, we also extracted the embeddings from the initial node embedding layer without loading any checkpoints.

### A.3.2 VISUALIZATION

Figure 10-13 show the t-SNE visualization results of node embeddings before and after training from GCN and SAGE with node features denoted as 'Operator w/ Attributes.' The orange points represent Younger's top 500 most frequently occurring operators. It can be observed that before training, the distribution of node embeddings is relatively concentrated and chaotic. After training, the distribution of embeddings representing high-frequency nodes selected and other low-frequency nodes from Younger was well distinguished. This indicates an uneven distribution of node quantities among different types, which introduces bias into the learning process of baseline models.

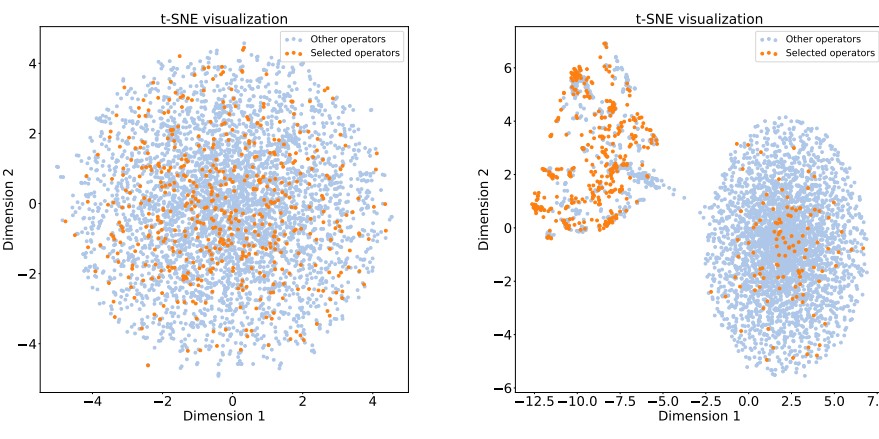

Figure 10: Node embeddings before training (GCN - Operator w/ Attributes)

Figure 11: Node embeddings after training (GCN - Operator w/ Attributes)

Figure 14-19 show the t-SNE visualization results of node embeddings before and after training from GCN, GAT, and SAGE with node features denoted as 'Operator w/ Attributes.' The orange points represent Younger's top 20 most frequently occurring operators. It can be seen that the distribution of node embeddings is relatively concentrated before training, while the distribution of all embeddings is uniform after training. This result indicates baseline models learned the features of different nodes well.

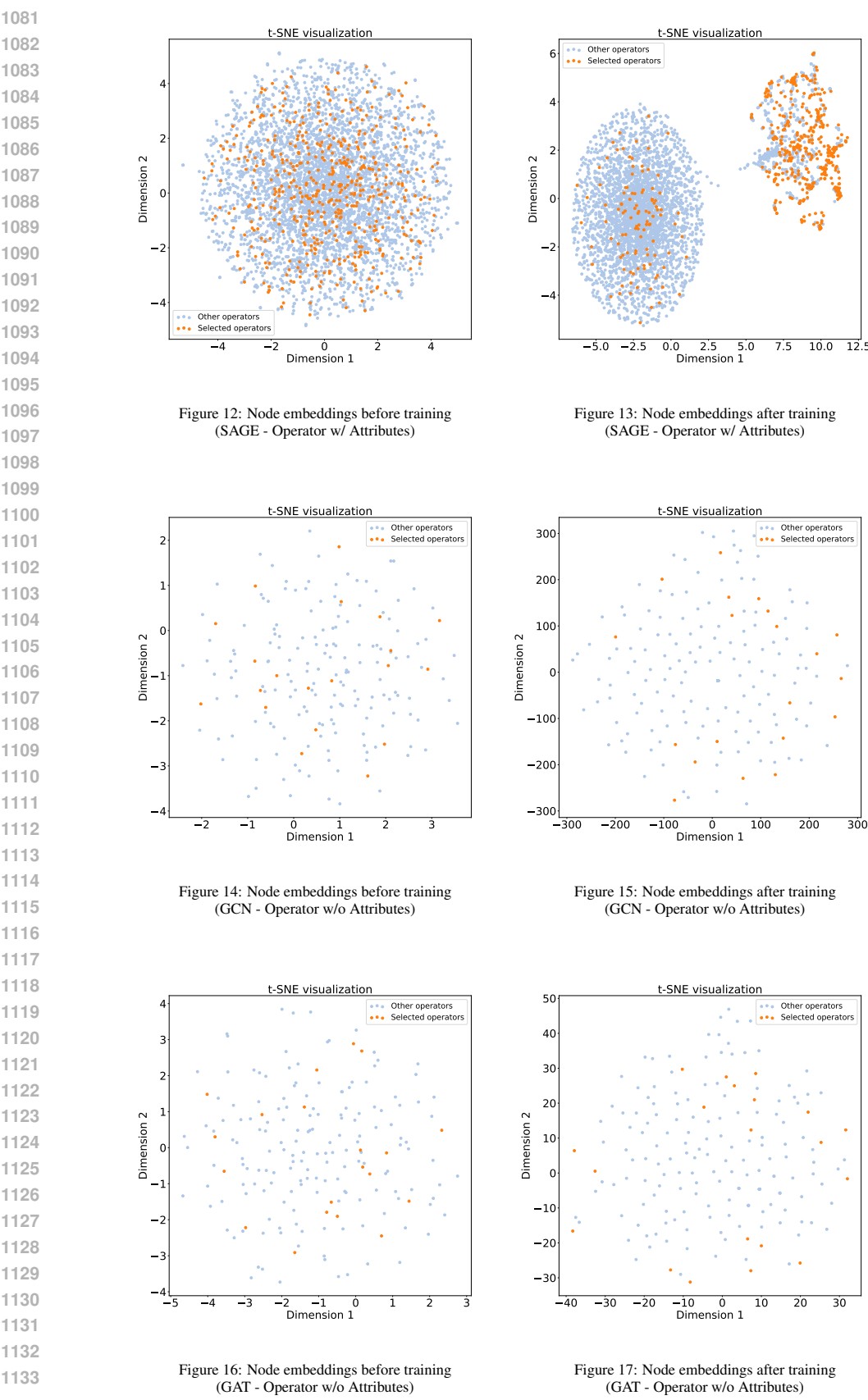

Figure 12: Node embeddings before training
(SAGE - Operator w/ Attributes)

Figure 13: Node embeddings after training
(SAGE - Operator w/ Attributes)

Figure 14: Node embeddings before training
(GCN - Operator w/o Attributes)

Figure 15: Node embeddings after training
(GCN - Operator w/o Attributes)

Figure 16: Node embeddings before training
(GAT - Operator w/o Attributes)

Figure 17: Node embeddings after training
(GAT - Operator w/o Attributes)

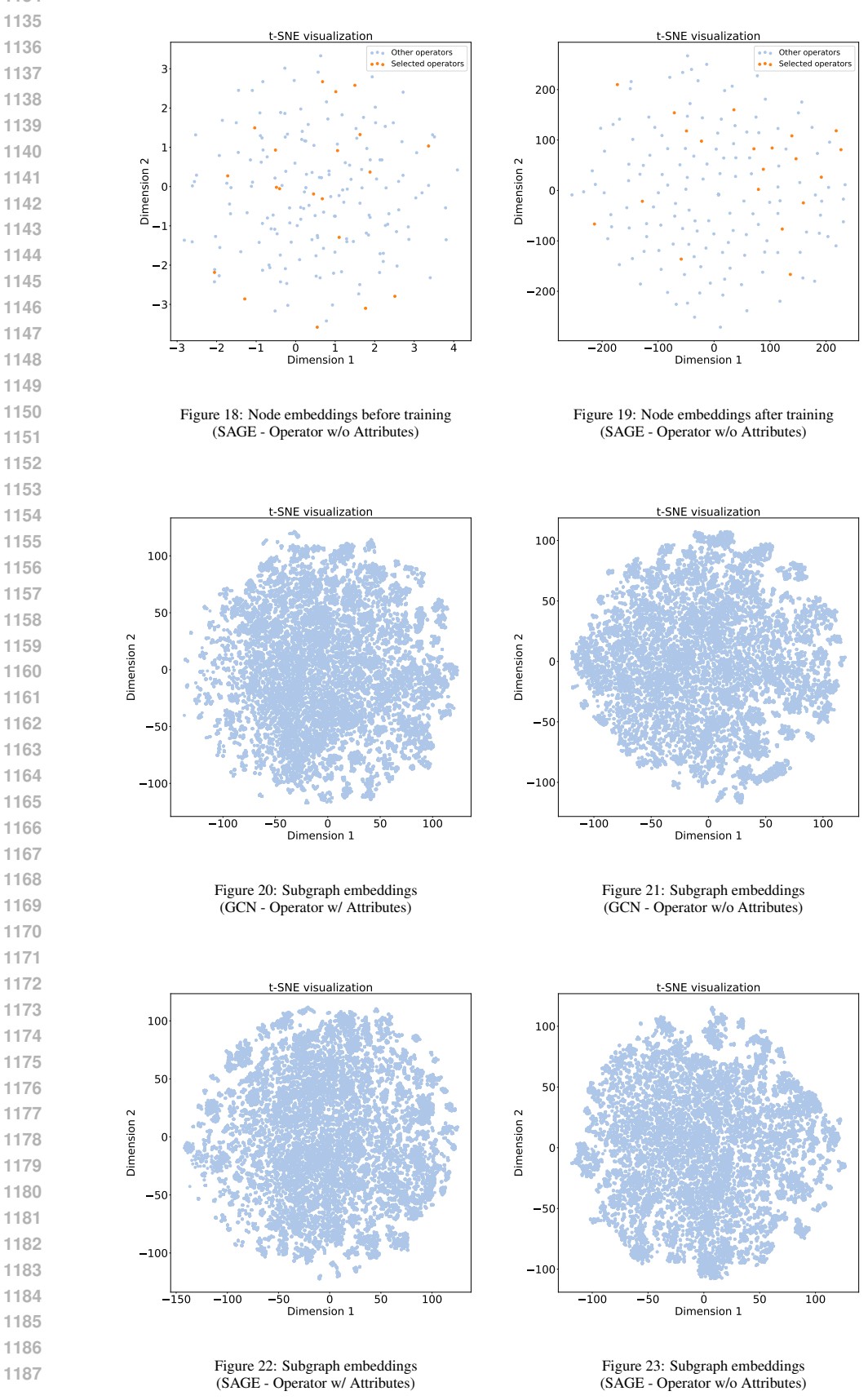

Figure 18: Node embeddings before training
(SAGE - Operator w/o Attributes)

Figure 19: Node embeddings after training
(SAGE - Operator w/o Attributes)

Figure 20: Subgraph embeddings
(GCN - Operator w/ Attributes)

Figure 21: Subgraph embeddings
(GCN - Operator w/o Attributes)

Figure 22: Subgraph embeddings
(SAGE - Operator w/ Attributes)

Figure 23: Subgraph embeddings
(SAGE - Operator w/o Attributes)

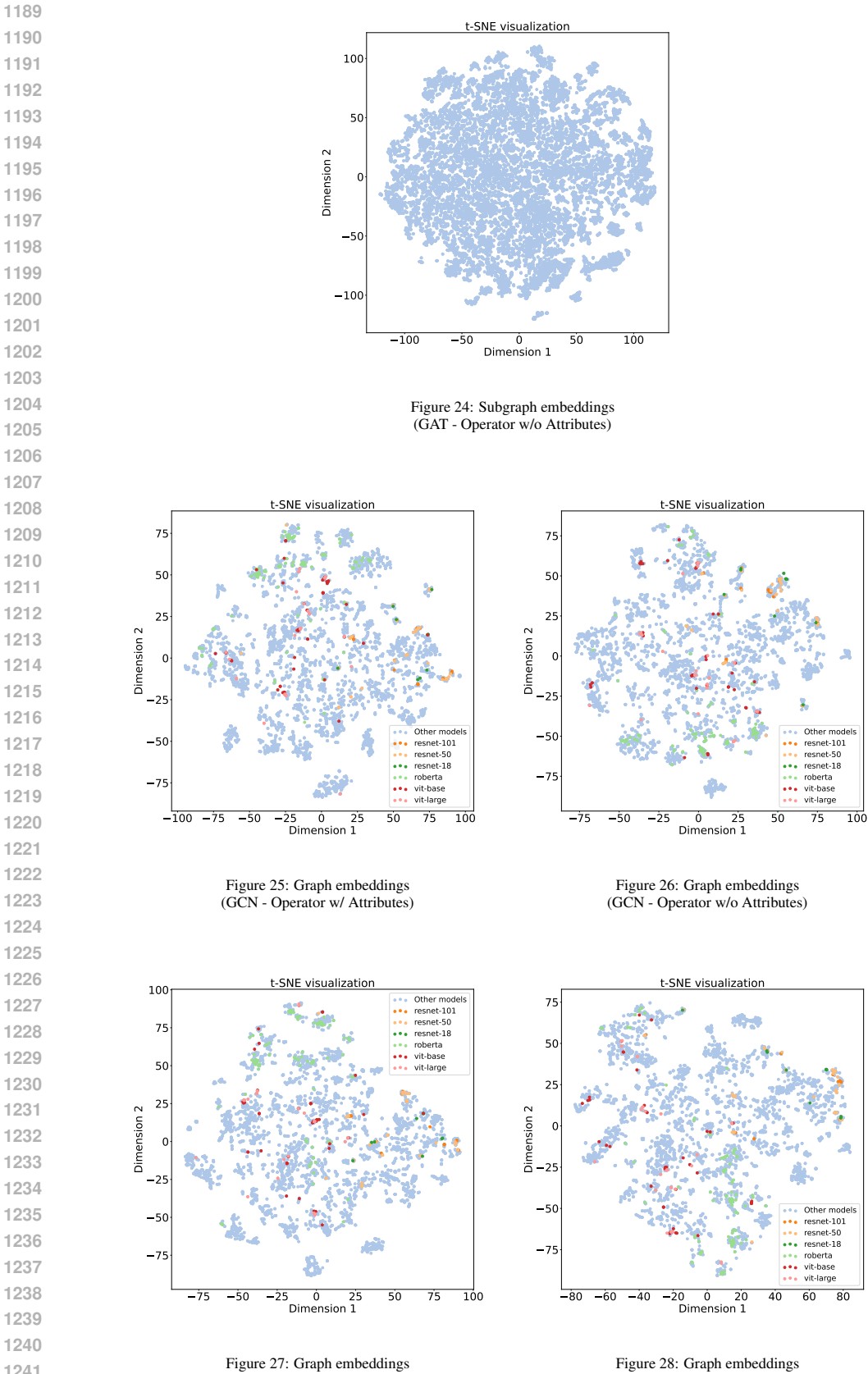

Figure 24: Subgraph embeddings
(GAT - Operator w/o Attributes)

Figure 25: Graph embeddings
(GCN - Operator w/ Attributes)

Figure 26: Graph embeddings
(GCN - Operator w/o Attributes)

Figure 27: Graph embeddings
(SAGE - Operator w/ Attributes)

Figure 28: Graph embeddings
(SAGE - Operator w/o Attributes)

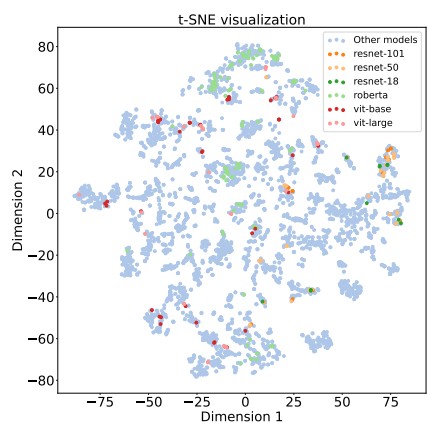

Figure 29: Graph embeddings
(SAGE - Operator w/o Attributes)

### A.4 SUBGRAPH EMBEDDING

#### A.4.1 CHECKPOINT SELECTION

To better illustrate the distribution of subgraphs in Younger in high-dimensional space, we selected checkpoints of baseline models according to the method from Section A.2.5. Then, we calculated the embeddings of these subgraphs using operators with attributes and those without attributes.

#### A.4.2 VISUALIZATION

Figure 20-24 show the t-SNE visualization results of all subgraph embeddings under the GCN, GAT, and SAGE models. Due to memory overflow in 'Operator w/ Attributes' of GAT, we only present the visualization of 'Operator w/o Attributes' about GAT. As can be seen, all three models have distinguished the embeddings of subgraphs well, but due to data bias, node embeddings were not learned particularly well. Therefore, the models only distinguished the embeddings of subgraphs well in a part of the spatial distribution (the boundary of the space). In addition, compared to 'Operator w/o Attributes,' 'Operator w/ Attributes' has a finer granularity in distinguishing subgraph embeddings, i.e., different clusters occupy less space.

### A.5 GRAPH EMBEDDING

#### A.5.1 OBTAINING METHOD

We obtain the graph embeddings by averaging the embeddings of all subgraphs in each DAG. Therefore, each baseline model can generate two types of graph embeddings: 'Operator w/Attributes' and 'Operator w/o Attributes.' However, due to memory overflow in 'Operator w/ Attributes' of GAT, we only present the visualization of 'Operator w/o Attributes' about GAT.

#### A.5.2 VISUALIZATION

Figure 25-29 show the t-SNE visualization results of all graph embeddings under the GCN, GAT, and SAGE models. We mark the embeddings of several commonly used models in figures in different colors. Several architectures have shown almost similar results. It can be seen that, on the one hand, the embeddings of DAGs based on the same architecture are very close or even overlap in the graph; for example, there are many points of the RoBERTa (Liu et al., 2019) and ViT (Dosovitskiy et al., 2020) architectures, which are Transformer-based (Vaswani et al., 2017) architectures, that are close in distance or overlap. On the other hand, it can be seen that the Younger dataset covers multiple common architectures well, indicating that Younger covers most of the neural network architectures

in the real world. In addition, the same architecture has multiple points of the same color in the figures, indicating that the dataset contains various variants of this type of architecture.

