# OpenReview forum: "Younger: The First Dataset for Artificial Intelligence-Generated Neural Network Architecture"
_ICLR.cc/2025/Conference — ICLR 2025 Conference Withdrawn Submission_

### Official Review · Reviewer_bp8h · 2024-10-29

**Soundness:** 2
**Presentation:** 1
**Contribution:** 1
**Rating:** 3
**Confidence:** 5

**Summary:**

This manuscript proposes Younger, a dataset comprising of neural network ONNX graph representations collected from various online repositories. Each ONNX graph is annotated with some kind of task performance. The goal of Younger is to fuel Artificial Intelligence-Generated Neural Network Architecture (AIGNNA) generation, in order to break the confines of pre-established search spaces. The manuscript consists of tables and figures tabulating/illustrating features of the Younger dataset, as well as some limited experiments on AIGNNA Exploration.

**Strengths:**

Strengths:
- The dataset consists of many diverse architectures from different tasks.
- Breaking the confines of predefined search spaces is a good step forward for NAS.

**Weaknesses:**

Weaknesses:
- Primary weakness of this paper is that its key contribution is not novel at all. While it is true that predefined search spaces pose a large problem for NAS, there are existing works that break the confines of predefined search spaces, enabling generalizable prediction across different micro, cell-based NAS search spaces [1] and even between micro and macro search spaces [2], using ONNX as the generalizable representation [3, 4, 5] or otherwise. This paper does not seem to recognize such existing works.
- AIGNNA in Section 4.3, the authors illustration is flawed for Fig. 7 left. You would not have a choice between 'Add' and 'ReLU' since ReLU is an activation applied to one single input, while 'Add' combines several inputs into one. A better choice would be to ask 'ReLU' or 'SiLU' or 'Add' vs. 'Concat'. Here still, there is existing work on generating new architectures outside of pre-existing search spaces [6, 7].
- Experimental results in Section 4.3.2 is not very convincing as you are focusing on testing different GNN designs (which govern the message passing rule, not necessarily graph feature design), while there are numerous neural predictors in the literature [8] which are better compared to. Also, ACC, F1, Prec. and Recall are generally not evaluation metrics for neural predictors; rather, rank correlation (Kendall's Tau or Spearman Rho) and regression error (L1) are more common [8].
- Overall presentation in the paper is very lacking. Table and Figure captions are too short and do not adequately convey enough information, while most figures (2 - 6) should be re-tweaked with larger fonts.

**Questions:**

Section 4.3.2: different tasks have different evaluation metrics and ranges for different datasets [9], so how do you handle that?
Section 4.3.3: Can you provide more information on exactly what this is, in terms of framing it through citations and potential tables/figures presenting the results? That would be a better use of page real estate than Figs 2-6 which are more appendix details.

References:

[1] Liu, Yuqiao, et al. "Bridge the gap between architecture spaces via a cross-domain predictor." Advances in Neural Information Processing Systems 35 (2022): 13355-13366.

[2] Mills, Keith G., et al. "Gennape: Towards generalized neural architecture performance estimators." Proceedings of the AAAI Conference on Artificial Intelligence. Vol. 37. No. 8. 2023.

[3] Yang, Yichen, et al. "Equality saturation for tensor graph superoptimization." Proceedings of Machine Learning and Systems 3 (2021): 255-268.

[4] Zhang, Chenhao, et al. "Towards better generalization for neural network-based sat solvers." Pacific-Asia Conference on Knowledge Discovery and Data Mining. Cham: Springer International Publishing, 2022.

[5] Mills, Keith G., et al. "Building Optimal Neural Architectures using Interpretable Knowledge." Proceedings of the IEEE/CVF Conference on Computer Vision and Pattern Recognition. 2024.

[6] Schrodi, Simon, et al. "Construction of hierarchical neural architecture search spaces based on context-free grammars." Advances in Neural Information Processing Systems 36 (2024).

[7] Salameh, Mohammad, et al. "AutoGO: automated computation graph optimization for neural network evolution." Advances in Neural Information Processing Systems 36 (2024).

[8] White, Colin, et al. "How powerful are performance predictors in neural architecture search?." Advances in Neural Information Processing Systems 34 (2021): 28454-28469.

[9] Mills, Keith G., et al. "Aio-p: Expanding neural performance predictors beyond image classification." Proceedings of the AAAI Conference on Artificial Intelligence. Vol. 37. No. 8. 2023.

---

### Official Review · Reviewer_8rBi · 2024-10-30

**Soundness:** 3
**Presentation:** 3
**Contribution:** 3
**Rating:** 6
**Confidence:** 3

**Summary:**

The paper presents a dataset for neural network architectures extracted from public model sources. Then convert these models to intermediate representation operators for further research purpose.  The dataset incorporates richer operator than existing neural architecture datasets, to facilitate more research in the field.

**Strengths:**

1. The paper presents a good contribution to the community by offering a dataset that can potentially faciliate the research in the neural architecture search field.

2. The dataset published can overcome challenges of previous related datasets in terms of operator scope and scale.

**Weaknesses:**

One complaint I have regarding this dataset comparing with existing NASbench dataset is that the proposed dataset doesn't seem to have the associated training performance on a standardized dataset.  In comparison, the NASbench dataset is evaluated on a standard task (Cifar10) on different settings.  Because of this, user can only perform unsupervised analysis like shown in figure 3-6.

**Questions:**

in the paper it is stated in line 284 that "less than 1% of these models represent heterogeneous and effective architectures. This notably low proportion of heterogeneous architectures highlights the limitations of current neural network design methods, both manual and NAS-based, in fostering architectural innovation".

How did the author decide whether an architecture is heterogeneous?  and what does it mean by effective architecture?   In addition, all architectures are from online sources should be mostly manually built right?    It is really surprising that out of all models available online, less than 1% are really valid.

---

### Official Review · Reviewer_eSNQ · 2024-11-03

**Soundness:** 3
**Presentation:** 3
**Contribution:** 3
**Rating:** 6
**Confidence:** 3

**Summary:**

This paper presents a novel dataset for AI-generated neural network architectures. Specifically, the construction process contains four core steps, i.e., retrieving NN models, converting the models to ONNX format, extracting DAGs from the ONNX models, and filtering out isomorphic DAGs to ensure the uniqueness of the architectures. Some experimental results support the statistical analysis and present some distributions of the whole dataset.

**Strengths:**

1) Constructing the dataset for neural architectures in the AI-generated manner is novel and interesting.
2) The paper is nicely written and well organized. The details for the dataset and construction processes are clearly presented.
3) The AI-generated architectures in different types are very helpful to several research directions.

**Weaknesses:**

1) More details in terms of the use of the proposed YOUNGER dataset are expected. For example, it is possible to provide at least one case using the YOUNGER dataset, such as performing some NAS algorithms to search for architectures in this dataset?
2) The experimental results are mainly focus on GNNs. However, it seems that there can be other types of architectures contained by YOUNGER. Could the authors provide additional experimental results in terms of other types of architectures beyond GNNs?
3) More details for the distribution of the performance of architectures are needed.

**Questions:**

Please see the weaknesses. Overall, this paper is interesting and the proposed YOUNGER dataset seems to be useful in several directions. Giving all these, I’d like to recommend the score 6 temporarily.

---

### Official Review · Reviewer_j9Uy · 2024-11-04

**Soundness:** 3
**Presentation:** 3
**Contribution:** 2
**Rating:** 5
**Confidence:** 4

**Summary:**

This study produced a dataset of 7k unique models, Younger, from 174k publicly available models and 30 tasks. After processing and filtering, architectures are stored as acyclic graphs based on ONNX definitions.  The study aims to automate architecture generation and refinement.  It offers a range of statistical analyses to illustrate the diversity of Younger.  Several experiments are also included to show the potential of Younger, especially as a benchmark for GNN.

**Strengths:**

The study is highly relevant and timely for NAS-related fields, with the potential to generate a high impact on the community.

The effort invested in this work seems quite substantial.

The writing and the presentation are clear and easy to follow.

**Weaknesses:**

The goal of this study is quite ambitious, allowing people to search for good architectures using Younger for a wide range of tasks.  The experimental section shall match that ambition, e.g. demonstrating the applicability and benefits of Younger on a wide range of tasks.  This would help illustrate the practical application of the dataset.
* Provide a concrete example, or case study showing how Younger could be used for a specific task like CIFAR-10 classification and ImageNet classification.
* Other possible cases can be performing time series prediction
* Or creating generative models.
* Or multimodal models for a specific task e.g. text-to-video or description or caption generation task.

---

Also the paper claims Younger is advantageous in comparison with benchmarks like DARTS, NAS-Bench-201.  Other than the difference between them, as shown in Table 1, how would they compare in a specific task, e.g. using different sets for the same task?  The study should show Younger's advantages clearly, e.g. leading to better performance, reducing search costs etc.
* Conduct a comparative experiment using Younger and a benchmark like NAS-Bench-201 on a common task, measuring metrics like search efficiency and final model performance.

---

Clearly state whether AIGNNA is a novel term introduced in this paper.  If not, provide the appropriate citation.

---

The code and the dataset link cannot be found in the submission.
* Provide direct links to the code repository and dataset, perhaps in a dedicated "Resources" section.

**Questions:**

See above.

---

### Note · Authors · 2024-11-28

**Comment:**

**Dear Reviewers, Program Chairs, and Fellow Researchers,**
﻿

We are writing to request the withdrawal of our submission titled "Younger: The First Dataset for Artificial Intelligence-Generated Neural Network Architecture." (Paper ID: 13877) from consideration at ICLR 2025.


First and foremost, we sincerely appreciate the thoughtful and detailed feedback provided by the reviewers and the time and effort they have invested in evaluating our work.
﻿

After carefully considering the reviews, we have identified significant misunderstandings regarding the scope and positioning of our work, which may have arisen due to limitations in the clarity of our presentation or other factors.
﻿

However, we would like to reiterate that our work introduces the Younger dataset and explores its potential to enable new directions in AI-generated neural network architectures (AIGNNA). Contrary to some interpretations in the reviews, this work was neither intentionally nor unintentionally designed to establish direct relevance to neural architecture search (NAS). Specifically, NAS methods rely on well-defined, constrained search spaces for optimization. In contrast, the vast scale and diversity of the search space provided by Younger inherently make it unsuitable for direct application in traditional NAS workflows.
﻿

The primary goal of our work is to foster innovation by breaking away from pre-established, manually designed search spaces and enabling exploration in a broader and more diverse architectural landscape. This is undoubtedly a challenging endeavor, but we hope that by proposing such a dataset, we can provide a foundation and resources for researchers to explore new frontiers. While benchmarks such as NAS-Bench-* address fundamental issues in traditional NAS, their scope and objectives differ significantly from those of Younger. The Younger dataset aims to catalyze research in entirely new directions.
﻿

Given the significant misunderstandings regarding our intentions and the need to better articulate the distinctions and contributions of our work, we have decided to withdraw the submission. This decision will allow us to refine our framework, strengthen our experiments, and more clearly present the unique opportunities Younger has enabled in future submissions.
﻿

We sincerely thank the reviewers and program committee for their constructive and detailed feedback, which will undoubtedly help us improve the quality of our work.
﻿

*Sincerely,*

*All Authors*

**Withdrawal Confirmation:**

I have read and agree with the venue's withdrawal policy on behalf of myself and my co-authors.